# Compilation of Load Spectrum of Loader Working Device and Application in Fatigue Life Prediction

**DOI:** 10.3390/s25175585

**Published:** 2025-09-07

**Authors:** Xiaohua Shi, Wenming Guo, Jiyang Wang, Gang Li, Hao Lu

**Affiliations:** 1School of Mechanical Engineering, Yanshan University, Qinhuangdao 066000, China; xhshi@ysu.edu.cn (X.S.); guowenming420@163.com (W.G.); 18865201299@163.com (J.W.); 19565388873@163.com (G.L.); 2XCMG Construction Machinery Co., Ltd., Xuzhou 221000, China; 3College of Electronic Information and Automation, Tianjin University of Science and Technology, Tianjin 300222, China

**Keywords:** load spectrum, fatigue life prediction, wheel loader

## Abstract

During the working process of the wheel loader, the repeated cycle of the shoveling and unloading process will produce an impact, so the loader is under a cyclic load for a long time, which leads to the frequent failure of its main parts. In this study, a new way of compiling the load spectrum of the loader’s working device and its application in fatigue life prediction is proposed. Through experimental data collection and preprocessing, the force of the cylinder block and hinge contact is corrected by mapping and inertia, which accurately reflects the actual force of the loader. The whole life cycle load spectrum is compiled by using the rainflow counting method and the extrapolation coefficient, and the test efficiency is optimized with the low-amplitude load omission method. By combining finite element analysis with material S-N curves using nCode DesignLife (version 11.1) and ANSYS Workbench frameworks (version 2024 R2), this research accurately predicts the fatigue life of the loader’s working unit and identifies key failure areas. The prediction results are consistent with the actual feedback data, and the accuracy of the method is verified.

## 1. Introduction

Wheel loaders are essential construction machinery primarily tasked with loading and transporting bulk materials such as rocks and soil. They are widely employed in diverse settings, including mining areas, ports, and construction sites. The operational loading cycle consists of four key phases: digging, hauling, dumping, and leveling [1,2]. During operation, the repeated cyclic process of loading and unloading generates significant shocks, subjecting the loader to prolonged cyclic loading. This often leads to frequent failures of critical components [3]. Typical failure modes of key structural parts in the loader’s working mechanism are illustrated in Figure 1.

The load spectrum serves as a fundamental input for the reliability design of mechanical components. It provides essential data for life cycle design, dynamic simulation, and finite element analysis, while also forming the basis for fatigue life assessment and accelerated testing [4,5,6,7,8,9,10,11]. The block program load spectrum was first proposed by Gassner in 1935 and applied to fatigue testing of aircraft structures [12]. To date, this programmed load spectrum remains widely used in fatigue load testing for various critical components. Its key advantage lies in its compatibility with the Palmgren–Miner linear damage accumulation rule [13] and its ease of application in fatigue testing machines. Jixin Wang et al. [14] presented a load extension-based nonparametric rainflow extrapolation for wheel loader semi-axle load spectrum compilation. Xuming Niu et al. [15] proposed a CMS compilation method for aero-engine maneuvering load by dividing and modeling typical mission segments. Xuetong Wang et al. [16] compiled a fixed frog nose rail load spectrum via aelliptic Gaussian kernel density, extrapolating short-term rainflow matrices.

Currently, most researchers have improved upon traditional methods to enhance the accuracy of life prediction. Jie Li et al. [17] proposed a novel model for predicting fatigue crack propagation life in welded structures by considering residual stress and load history effects, significantly improving prediction accuracy. Penghui Wang et al. [18] introduced a new methodology for compiling load spectra for hydraulic excavators, enabling more efficient acceleration of fatigue testing processes. Chaotao Liu et al. [19] assessed the residual life of metro bogies by developing full life cycle stress spectra, providing critical insights for maintenance strategy optimization. Xizhou Yang et al. [20] employed a parametric approach to extrapolate and compile one-dimensional load spectra for automotive rear suspension lower control arms. Additionally, they modified the Corten–Dolan model, which established an improved fatigue life prediction framework.

Fatigue failure at the weld of the wheel loader working device needs to be accurately compiled and used for fatigue life prediction to solve the failure problem. This research mainly includes the preparation and inertial correction of the load spectrum of the working device of the wheel loader, and the fatigue life prediction based on the cumulative damage theory.

## 2. Load Spectrum Collection and Preprocessing of Loader Working Device

The test prototype, an XC968 loader (XCMG Construction Machinery Co., Ltd., Xuzhou 221000, China), used the cylinder displacement sensor and the cylinder pressure sensor to measure the movement information of the cylinder used by the loader working device in the working process, which can help to easily derive the position of the working device. The cylinder displacement sensor and the cylinder pressure sensor are shown in Figure 2.

The pin force transducer shown in Figure 3 was used to measure the pin force at the lower pin of the boom in the loader working device, to verify the bucket load derivation process.

The sensor-acquired data exhibited abrupt anomalous values during signal analysis. By employing a differential gradient algorithm and dynamic thresholds for spike identification, an automated detection and elimination process for abnormal spike data points was implemented within the nCode GlyphWorks [21] software platform (version 11.1). The SpikeDetection module uses the differential gradient algorithm and dynamic threshold to eliminate the abnormal spike data and uses the semi-sine curve interpolation method to connect and refine the data repair process. At the same time, a low-pass filter with a cut-off frequency of 20 Hz is used to filter out the high-frequency noise components. The data before and after processing are shown in Figure 4.

## 3. Compilation of Equivalent Load Spectrum of Loader Bench Test Cylinder

The fatigue testing of the working mechanism was conducted in the laboratory through cylinder loading with a fixed posture. However, the collected cylinder pressure data reflected the continuously changing postures of the working mechanism. Therefore, it was necessary to map the variable-posture cylinder pressure to the bucket tip at a fixed bucket posture [22]. Subsequently, the loading spectrum generated at the bucket tip was remapped back to the cylinder according to the test bench posture, ultimately forming an equivalent cylinder loading spectrum that accurately represented the actual loading process.

### 3.1. Derivation and Verification of Cylinder Force to Bucket Tip Force

As shown in Figure 5, the external force on the rocker arm is described as follows: hinge force $F_{Dx}$ and $F_{Dy}$ at the hinge point D, the tipper cylinder force $F_{E}$ at the hinge point E, and bucket tie rod force $F_{C}$ at hinge point C.

As shown in Figure 6, the line connecting the hinge point G and the A point is described as the tangential axis t, which is perpendicular to the straight line. The direction of GA is the normal axis n. The external forces applied to the boom portion can be obtained from Figure 6 as follows: hinge point forces $F_{Gn}$, $F_{Gt}$ at hinge point G, boom cylinder force $F_{I}$ at hinge point I, hinge point forces $F_{Dx}$ and $F_{Dy}$ at hinge point D, and hinge point forces $F_{At}$ and $F_{An}$ at hinge point A.

At hinge point D of the boom, it is necessary to convert the coordinate system, and the conversion relationship is shown as follows:(1)$$\left\{\begin{array}{l} F_{Dt}=F_{Dx}\cos\alpha -F_{Dy}\sin\alpha \\ F_{Dn}=F_{Dx}\sin\alpha +F_{Dy}\cos\alpha \end{array}\right.$$
where $\alpha =90{}^{\circ}-\beta_{3}-\beta_{4}$, $\beta_{3}$ is the angle between the AB line and the vertical direction, and $\beta_{4}$ is the angle between the AB line and the vertical direction.

The following equations can be obtained by taking moment balancing for several hinge points in the boom and rocker:(2)$$\left\{\begin{array}{c} F_{E}\left(\sin\beta_{1}y_{DE}+\cos\beta_{1}x_{DE}\right)-F_{C}\left(\sin\beta_{2}y_{DC}+\cos\beta_{2}x_{DC}\right)=0\\ F_{In}t_{GI}+F_{It}n_{GI}+F_{Dn}t_{GD}-F_{Dt}n_{GD}-F_{An}t_{GA}+g_{t}n_{Gg}-g_{n}t_{Gg}=0\\ F_{Gn}t_{GA}-F_{In}t_{IA}+F_{It}n_{IA}-F_{Dt}n_{AD}-F_{Dn}t_{AD}+g_{t}n_{Ag}+g_{n}t_{Ag}=0\\ F_{Gn}t_{GI}-F_{Gt}n_{GI}+F_{Dn}t_{ID}-F_{Dt}n_{ID}+F_{At}n_{IA}-F_{An}t_{IA}+g_{n}t_{Ig}-g_{t}n_{Ig}=0 \end{array}\right.$$

In addition, the following equation can be obtained for the horizontal and vertical force balance of the boom:(3)$$\left\{\begin{array}{c} F_{Dx}-F_{C}\sin\beta_{2}-F_{E}\sin\beta_{1}=0\\ F_{Dy}-F_{C}\cos\beta_{2}-F_{E}\cos\beta_{1}=0 \end{array}\right.$$

The normal forces $F_{An}$ and tangential force $F_{At}$ at the hinge A can be solved with Equations (2) and (3):(4)$$F_{An}=\frac{F_{In}t_{GI}+F_{It}n_{GI}+F_{Dn}t_{GD}-F_{Dt}n_{GD}+g_{t}n_{Gg}-g_{n}t_{Gg}}{t_{GA}}$$

Substituting Equation (4) into Equation (2) can obtain the following results:(5)$$F_{At}=\frac{1}{2}\left(\frac{-F_{Gn}t_{GD}+F_{Dn}t_{ID}-F_{Dt}n_{ID}+F_{An}t_{IA}-F_{Dt}n_{IA}}{n_{GI}}-\Psi \right)$$
where(6)$$\Psi =\frac{F_{Gn}t_{GD}-F_{In}t_{GD}+F_{It}n_{ID}+F_{An}t_{AD}}{n_{GD}}$$
where $t_{IA}$ and $n_{IA}$ are expressed as the tangential distance and normal distance between Point I and Point A, respectively; $t_{ID}$ and $n_{ID}$ are expressed as the tangential distance and normal distance between point I and Point D, respectively; and $g_{t}$ and $g_{n}$ are the tangential force and normal force of point g. The tangential force and normal force of the hinge point at A can be calculated from the equation, and the direction of the hinge point force at A can be further obtained.

The cylinder forces during variable-posture loading operations were mapped to the bucket tip at a fixed posture based on the bucket configuration in fatigue bench tests. Initially, according to the mechanical relationship between cylinder forces and the lower hinge point (connecting the bucket and boom), a bending moment calculation point was selected on the boom’s neutral plane. Using the bending moment section equivalence method [23], the cylinder forces were first mapped to the hinge point with inertial force compensation, then transferred to the bucket tip based on the kinematic mapping relationship between the hinge point and bucket tip [24]. Following the fatigue damage equivalence principle for critical boom locations, the transformation from bucket tip and linkage forces to cylinder forces under fixed-posture conditions was defined as forward mapping, while the conversion from cylinder forces to bucket tip forces under variable-posture conditions was defined as reverse mapping. Both of these mapping processes were implemented in MATLAB (version R2022b) to obtain the equivalent bucket tip load spectrum; the fatigue test loading postures are shown in Figure 7.

During cylinder–hinge point mapping, to verify the accuracy of the mapping relationship, it is necessary to make a comparison during the mapping process. The hinge force between the bucket and the boom and the hinge force between the bucket and the connecting rod were measured by the pin sensor and the connecting rod sensor and were compared with the value mapped to the measuring point by the cylinder force. To facilitate verification, the cylinder force was mapped to the hinge point position under the conditions of shoveling 3 tons, 5 tons, and 7 tons of heavy blocks and maximum rising force. Figure 8 shows a comparison between the mapping value and the measured data. The results show that the maximum error between the mapping value of the hinge point force between the bucket and the boom and the measured value is not more than 8%, the average error is 6.81%, the root mean square error is 0.52 tons, and 80% of the data have a relative error of less than 10%. For the hinge force between the bucket and the linkage, the maximum error between the mapped value and the measured value is less than 6%, the average error is 2.47%, the root mean square error is 1.36 tons, and 90% of the data have a relative error of less than 10%. The above analysis results verify the correctness of the mapping relationship obtained in the cylinder–hinge experiment.

In the loader shoveling process, when the loading arm was lifted, the cylinder pressure signal not only contained the real resistance of material excavation but also mixed with the inertia component caused by the acceleration change; the inertia force frequency caused by the impact was higher than 20 Hz, and the pretreatment stage would eliminate this impact. Therefore, to avoid large prediction errors in fatigue life, it is crucial to correct the force mapped to the hinge point, especially in the rapid acceleration or emergency stop phase, because the inertial force might even dominate.

### 3.2. Inertia Correction of the Hinge Point Force of the Loader Working Device

To investigate the inertial forces at the hinge joint between the boom and bucket, a dynamic model of the XC968 working mechanism was developed [25]. For large-mass, high-speed working mechanisms, neglecting inertial effects would fail to capture their true mechanical characteristics. Therefore, each component of the loader’s working mechanism was treated with lumped mass modeling, and the dynamic analysis was performed using equivalent force systems. The acceleration at each lumped mass point was calculated using the five-point fourth-order central difference method. Taking the x-coordinate as an example, the five-point fourth-order difference scheme for calculating lumped mass point acceleration is given in Equation (7):(7)$$a_{i}(x)=\frac{3\left(-x_{i-2}+12x_{i-1}-26x_{i}+18x_{i+1}-2x_{i+2}\right)}{3{\left(t_{i+2}-t_{i-2}\right)}^{2}}$$
where $x_{i}$ is the abscissa of the mass lump point at time i, and $a_{i}(x)$ is the transverse acceleration of the mass lump point at time i.

After obtaining the acceleration of the mass lump point, the equivalent inertial force at the mass lump point is calculated according to Newton’s second law:(8)$$m\left\{\begin{array}{l} a(x)\\ a(y) \end{array}\right\}=\left\{\begin{array}{l} F(x)\\ F(y) \end{array}\right\}$$
where F(x), F(y) is the transverse inertial force and longitudinal inertial force of the mass lump point; a(x), a(y) is the lateral acceleration and longitudinal acceleration of the mass lump point, respectively; and m is the mass of the mass lump point.

The motion process of the loader working device was simulated, the displacement, velocity, acceleration, and other motion parameters of each structural part and the hinge point were collected by the data acquisition instrument. Based on the existing data, the simulation shown in Figure 9 was performed in Adams [26], and the velocity and acceleration data at the junction between the boom and the bucket model were collected.

By intercepting three complete operation cycles to analyze the movement characteristics of the loader working device, the acceleration obtained from the hinge point simulation of the boom and bucket was compared with the hinge acceleration used in the dynamic model in the horizontal direction, and the results are shown in Figure 10.

From the comparison chart, it can be seen that the theoretically derived acceleration value is more accurate when the impact is large. In addition, the variation trend of the velocity and acceleration of the boom and bucket hinge joints in the horizontal and vertical directions was collected using simulation. The velocity and acceleration curves are shown in Figure 11 and Figure 12.

From the simulation of the hinge velocity and acceleration curves, it could be seen that the velocity of the hinge between the bucket and the boom changed periodically. The excessively long intermittent time of the periodic shoveling operation was removed, and then the remaining periodic operation cycle was spliced through the displacement data of the cylinder in the shoveling process. The inertia force of the shoveling hinge point was then reproduced using the dynamic model, and the hinge point force of the bucket tip was mapped from the cylinder to carry out inertia correction to the hinge point force containing inertia, where the hinge point of the first two cycles of the curve correction was aligned, as shown in Figure 13.

According to the relationship between the bucket tip and the hinge point under the boom, the corrected hinge point force was mapped to the bucket tip of the loader, and the equivalent bucket tip load spectrum of the working device under the fixed attitude of the bench test was obtained. The results in Figure 14 show that the acquisition of the equivalent bucket tip load spectrum laid a foundation for the acquisition of the cylinder loading spectrum during the bench test.

### 3.3. Extrapolation of Load Spectrum and Compilation of Cylinder Loading Spectrum

The rainflow counting method, which is widely recognized in engineering applications, was employed to obtain the rainflow matrix [27].

Using the direct nonparametric rainflow extrapolation method, with an extrapolation coefficient of 10 [14], the load spectrum data of the bucket tip are extrapolated to derive a two-dimensional stress spectrum of 64 × 64, and the shape of the rainflow matrix shown in Figure 15 is obtained.

The two-dimensional load spectrum was converted into a one-dimensional load spectrum using the fluctuating center coordinate transformation method [28]. The resulting one-dimensional spectrum comprised eight levels, with variable-amplitude loading applied at specific mean values. The transformation process involved calculating, for each load level, the sum of the products of mean load values and their corresponding cumulative frequencies from the rainflow two-dimensional spectrum, divided by the total frequency count of all loads in the rainflow matrix. The specific calculation method is shown in Equation (9).(9)$$X_{m}=\frac{\sum X_{mi}\times P_{i}}{\sum P_{i}}$$
where $X_{m}$ represents the average load (fluctuation center), $X_{mi}$ represents the average load of each level, $P_{i}$ represents the frequency corresponding to the average load of each level, and $\sum P_{i}$ represents the total load frequency contained in the rainflow matrix.

Using the above method, the time series of the equivalent bucket tip load after inertia correction was processed, and the eight-level load spectrum was obtained, as shown in Table 1.

To speed up the test process, the low-amplitude load elision method [29,30] was used to remove small loads below the fatigue limit of 60%. By comparison, the predicted value of the deleted life expectancy was in good agreement with the original data. Based on the conservative principle of load compression, the test period was significantly shortened after removing the first three low-amplitude loads, as shown in Figure 16.

The loader carried out the bench test in a fixed attitude in the laboratory and applied the load to the working device through the rocker arm and boom cylinder, forming a force equivalent to the actual shoveling process. According to the equivalent fatigue damage of the fatigue concern point on the boom, the bucket tip equivalent load spectrum was mapped to the equivalent cylinder loading spectrum in combination with the connecting rod force, and the one-dimensional cylinder loading spectrum of the boom and rocker arm is shown in Figure 17 and Figure 18. The compilation of one-dimensional cylinder loading spectrum provided data support for the fatigue life prediction research of loader operating equipment.

## 4. Fatigue Simulation and Life Analysis of Loader Working Device

After obtaining the load spectrum of the working mechanism, it was necessary to acquire the corresponding S-N curve and combine it with the cumulative fatigue damage theory to predict the fatigue life of the working mechanism. However, determining the S-N curve for the entire working mechanism was challenging to achieve. Therefore, for vulnerable components of the working mechanism, such as the boom and welding joints of the rocker arm, the S-N curve was measured and subsequently used for fatigue life prediction of the working mechanism. The fatigue life prediction was then conducted based on the Palmgren–Miner rule.

### 4.1. S-N Curve of Weldments

By comparing with the weld structure of the working device, the shape of the welded part was finally selected as a T-shape, as shown in Figure 19. The material of this component is the same as the welding of the working device, and the welding method is the same. The static mechanical properties and performance characteristics of the material are shown in Table 2.

The fatigue test of the welded specimen was carried out on the GPS-200 high-frequency fatigue test system shown in Figure 20, and the stress ratio was 0.1. The amplitude of the testing machine was calculated, and then the average load was calculated according to the stress ratio. Finally, according to the calculation formula of tensile stress, the stress amplitude was obtained:(10)$$\sigma =\frac{F_{m}}{S}=\frac{F_{m}}{bh}$$
where b is the width of the specimen, h is the thickness of the specimen, $F_{m}$ is the load amplitude, and σ is the stress amplitude.

The fatigue testing machine determined the fatigue crack of the weldment through the change in loading frequency. The fatigue crack occurred in the weldment, and the fatigue tester would automatically stop. The magnetic trace at the crack of the sample is shown in Figure 21 and was obtained by using magnetic particle detection technology.

In the test, the number of cycles (N) of the sample when fatigue fracture occurs under different stress amplitudes (S) was recorded; the data are shown in Table 3, and the curve was formed by fitting the data points with the logarithmic coordinate system. The curve intuitively reflects the exponential increase in the fatigue life of the material with the decrease in stress amplitude and can be mathematically characterized using empirical formulas.

For materials with fatigue limits, the S-N curve will show a horizontal trend in the low stress level region, and the average value of the fatigue limit under each stress level is determined with multiple measurements of the fatigue limit under different stress levels. The average value of the fatigue limit is fitted to obtain the S-N curve as shown in Figure 22. The scatters in the graph represent the raw data recorded during the experiment.

### 4.2. Fatigue Life Prediction of Working Devices

The core hypothesis of the Palmgren–Miner rule [31] is that fatigue damage caused by different stress levels is mutually independent. In other words, identical loads will inflict equivalent damage to the material regardless of its current condition. For mechanical components subjected to multi-level alternating loads (with a total of m stress levels), the energy absorption value $W_{n}$ under the nth-level stress condition exhibits a deterministic functional relationship with the corresponding number of cycles at that stress level, as expressed in Equation (11).(11)$$\frac{W_{n}}{W}=\frac{n_{n}}{N_{n}}$$

According to Miner’s rule, Equation (12) can be obtained:(12)$$W_{1}+W_{2}+W_{3}+\cdots W_{m}=W$$

By simultaneously solving Equations (11) and (12), the final form of Miner’s criterion characterizing material fatigue behavior is derived as Equation (13):(13)$$D=\sum_{i=1}^{m}\frac{n_{i}}{N_{i}}=1$$
where W represents the energy limit value, and $N_{n}$ indicates the total number of cycles before the specimen undergoes fatigue failure at the nth stress level. When D=1, the structure will experience fatigue failure.

Through the collaborative application framework of nCode DesignLife and ANSYS Workbench [32], the finite element model and its static/transient dynamic analysis results could be transferred to the fatigue analysis module that employs the Palmgren–Miner theory. Meanwhile, in combination with the obtained load–time history (one-dimensional loading spectrum of the equivalent cylinder) and the material S-N curve, nCode achieved the visualization of fatigue damage distribution calculation and life prediction. The analysis process is shown in Figure 23.

The finite element model and cylinder loading spectrum were fed into the fatigue correlation module, and fatigue analysis parameters, such as material properties, stress combinations, and average stress correction methods, were defined in the nCode environment. After completing the above steps, it is possible to start calculating the fatigue life. The nCode DesignLife software is also very convenient for displaying and visualizing the results, such as fatigue life contours and node cycles, as shown in Figure 24.

The fatigue life distribution contour of the structure could be used to preliminarily determine the fatigue failure site, and the number of cycles of each node could be viewed in the DataValuesDisplys1 interface. After analysis, the minimum fatigue life cycles of the boom structure were 3.064 × 10^4^ times, and the critical area was located at the transitional connection between the lower wing plate and the lower boom plate. If the working conditions were based on the annual working cycle of 300 days and the average daily continuous operation for 8 h, and combined with the input load spectrum data for conversion, the actual service life evaluation results of the loading boom could be obtained:(14)$$T=\frac{N\times t}{60\times 60\times h\times d}=\frac{3.064\times 10^{4}\times 1850}{60\times 60\times 8\times 300}=6.56\;(\mathrm{Year})$$
where d is the number of working days per year, h is the average working time per day, and t is the simulation time.

In the same way, the fatigue life of the rocker arm was predicted, and the fatigue life distribution contour is shown in Figure 25.

According to the observation results of the fatigue life distribution map, the areas prone to fatigue failure of the rocker structure are mainly concentrated in the lower wing plate position. The lowest number of cycles was also 3.752 × 10^4^ times. Based on the average annual operating parameters of construction machinery, calculated according to 300 working days per year and 8 h of operation per day, combined with the input load spectrum data, the rocker life prediction model as shown in Equation (15) could be established:(15)$$T=\frac{N\times t}{60\times 60\times h\times d}=\frac{3.752\times 10^{4}\times 1850}{60\times 60\times 8\times 300}=8.03\;(\mathrm{Year})$$

Based on the statistical analysis of the fault cases during the service of the loader and the analysis of user return visits, the fatigue failure time of the welds of the boom and rocker arm of the equipment is mainly concentrated in the range of 5–10 years. The results of this engineering measurement are highly consistent with the results of the previous multi-condition simulation analysis, which verifies the engineering applicability of the simulation model to the fatigue life prediction of the structure.

## 5. Discussion

As described in this article, a systematic study was carried out on the preparation of the load spectrum of the loader working device and its application in fatigue life prediction. Through experimental collection and pretreatment, the load spectrum of the loader working device was successfully compiled and applied to fatigue life prediction. The research results showed that the prepared load spectrum can accurately reflect the actual force of the working device and provide reliable data support for fatigue life prediction. In the process of preparing the load spectrum, its accuracy was improved through the mapping and inertial correction of the cylinder force and the hinge point force. In addition, the test efficiency was significantly improved by optimizing the test cycle with the low-amplitude load omission method.

In terms of fatigue life prediction, through the collaborative application framework of nCode DesignLife and ANSYS Workbench, combined with finite element analysis and material S-N curve, the accurate prediction of the fatigue life of the working device was realized. The prediction results show that the fatigue failure parts of the boom and rocker arm are mainly concentrated in the connection parts of the lower wing plate and the transition, which are highly consistent with the actual service situation. This result not only verified the accuracy of the simulation model but also provided a reference for the design optimization and reliability improvement in the loader’s working device.

However, it was also found that some areas need further improvement. For example, in the process of preparing the load spectrum, although the accuracy of the load spectrum was improved by inertial correction, there were still certain errors in the calculation of inertial forces under complex working conditions. In addition, in the prediction of fatigue life, although the simulation results were consistent with the actual service conditions, the prediction of fatigue life under extreme operating conditions still lacks validation. Future research can optimize the calculation method of inertial force, verify the fatigue life prediction under extreme working conditions, and use sensor fusion to collect more data on actual working conditions. Combined with artificial intelligence, the accuracy and reliability of fatigue life prediction can be more effectively improved.

## 6. Conclusions

Through experimental data collection and preprocessing, the load spectrum of the whole life cycle of the loader working device was systematically compiled in this study, and the test efficiency was significantly improved through the optimization of low-amplitude loads; then, the load spectrum was applied to fatigue life prediction. The results show that the specifications achieved with inertial correction of cylinder force and hinge point force mapping can accurately reflect the actual force acting on the working device.

For fatigue life prediction, the integration of nCode DesignLife and ANSYS Workbench, combined with finite element analysis and material S-N curves, enabled precise predictions. The results reveal that fatigue failures in the boom and rocker arm predominantly occur at the lower wing plate connections and transition zones, aligning closely with real-world service observations. Furthermore, statistical analysis of failure cases and end-user feedback validate the engineering applicability of the simulation model for structural fatigue life prediction.

In summary, through the collection and preprocessing of experimental data, the load spectrum of the loader working device was established. Then, through the integration of the rainflow counting method and inertial correction technology, the full life cycle load spectrum of the loader working device was successfully extrapolated and compiled. Using the collaborative simulation framework of nCode DesignLife and ANSYS Workbench, it was predicted that the fatigue failure was mainly concentrated in the transitional connection area between the boom and the lower wing of the rocker arm, which is consistent with the actual service data. The research results provide a theoretical basis for optimizing the structural design of the loader and improving its reliability.

## Figures and Tables

**Figure 1 sensors-25-05585-f001:**
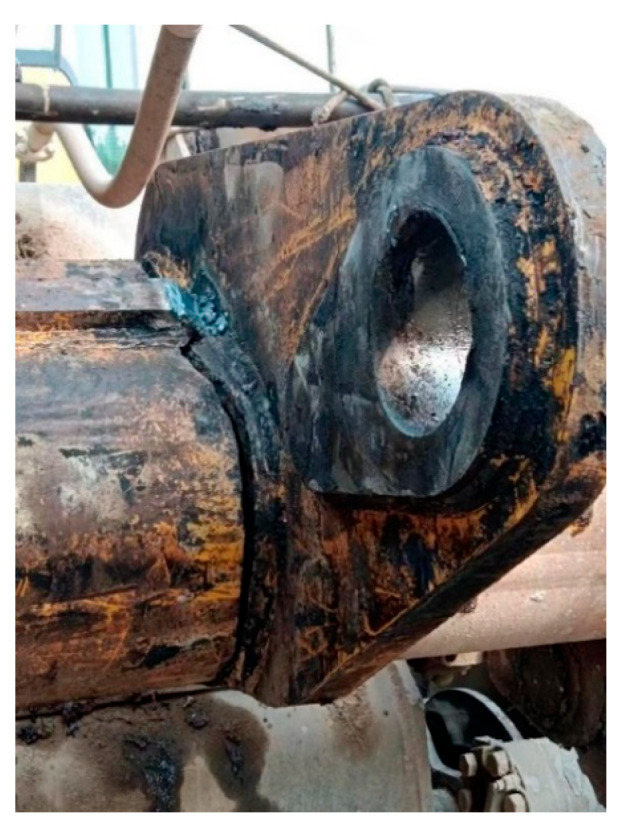
Failed parts of the loader working device.

**Figure 2 sensors-25-05585-f002:**
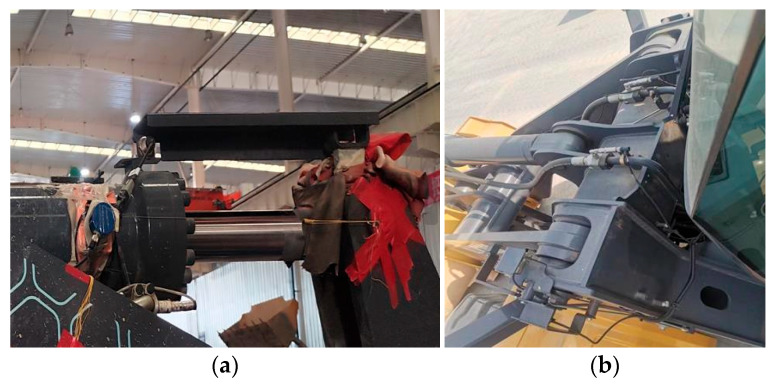
Sensor installation location at the cylinder: (**a**) installation position of cylinder displacement sensor; (**b**) installation location of cylinder pressure sensor.

**Figure 3 sensors-25-05585-f003:**
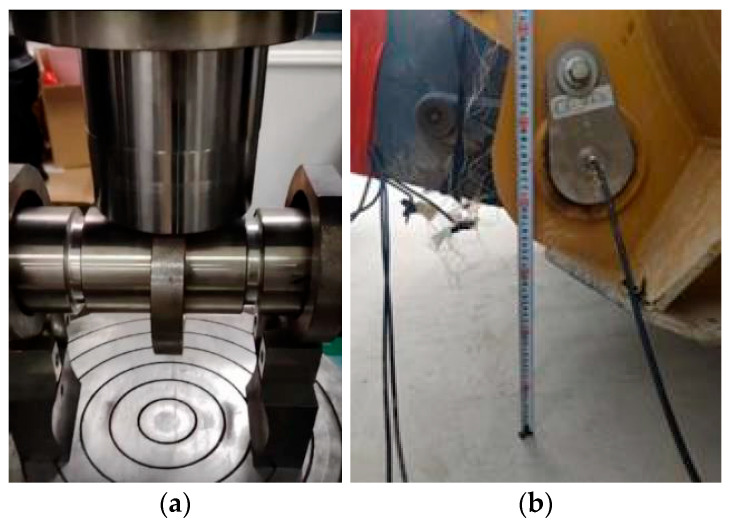
Pin sensor used for loader measurements: (**a**) pin sensor calibration process; (**b**) pin sensor installation location.

**Figure 4 sensors-25-05585-f004:**
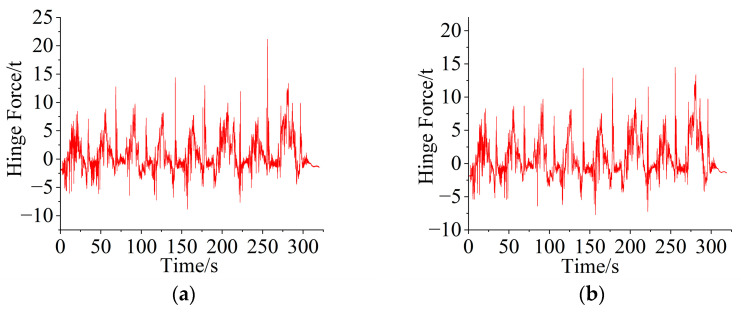
Comparison of data before and after preprocessing: (**a**) preprocessing data; (**b**) post-processing data.

**Figure 5 sensors-25-05585-f005:**
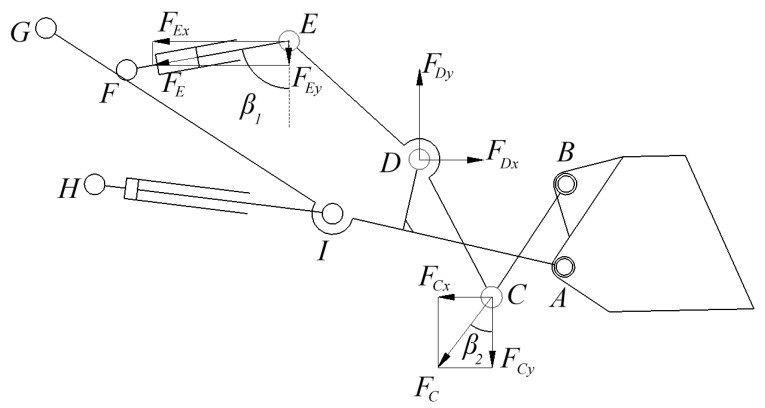
Rocker arm force analysis.

**Figure 6 sensors-25-05585-f006:**
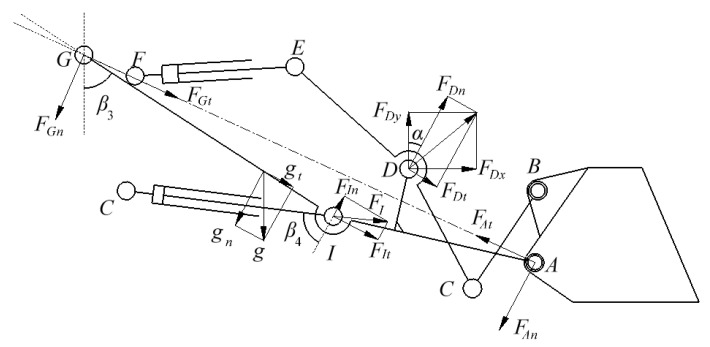
Boom force analysis.

**Figure 7 sensors-25-05585-f007:**
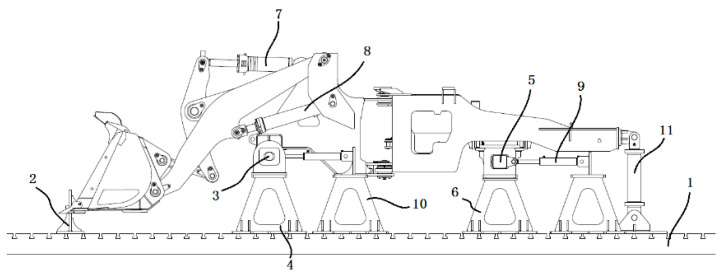
Fatigue bench test attitude: 1—T-stage, 2—bucket mount, 3—front rigid axle, 4—front rigid axle bearing, 5—rear rigid axle, 6—rear rigid axle bearing, 7—dump bucket cylinder, 8—boom cylinder, 9—traction cylinder, 10—traction cylinder bearing, 11—tail locking device.

**Figure 8 sensors-25-05585-f008:**
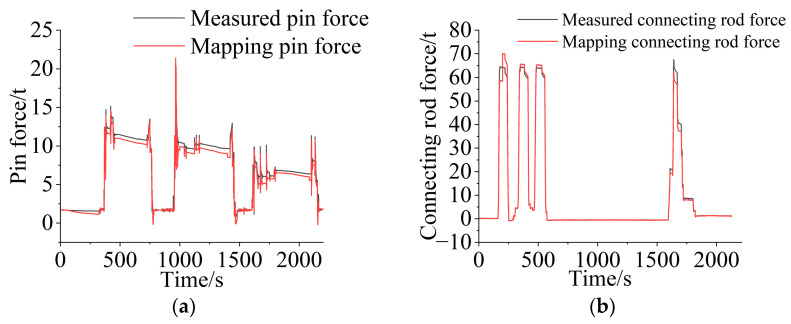
Derivation process verification: (**a**) measured value and mapping value of the pin axial force under heavy block conditions; (**b**) measured value and mapping value of the connecting rod force under the rising force condition.

**Figure 9 sensors-25-05585-f009:**
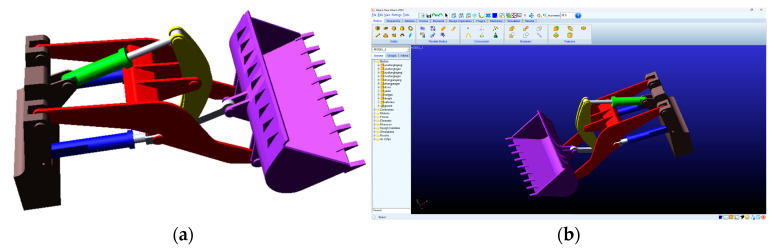
The simulation in Adams: (**a**) the model of the working device used in the simulation; (**b**) the simulation process.

**Figure 10 sensors-25-05585-f010:**
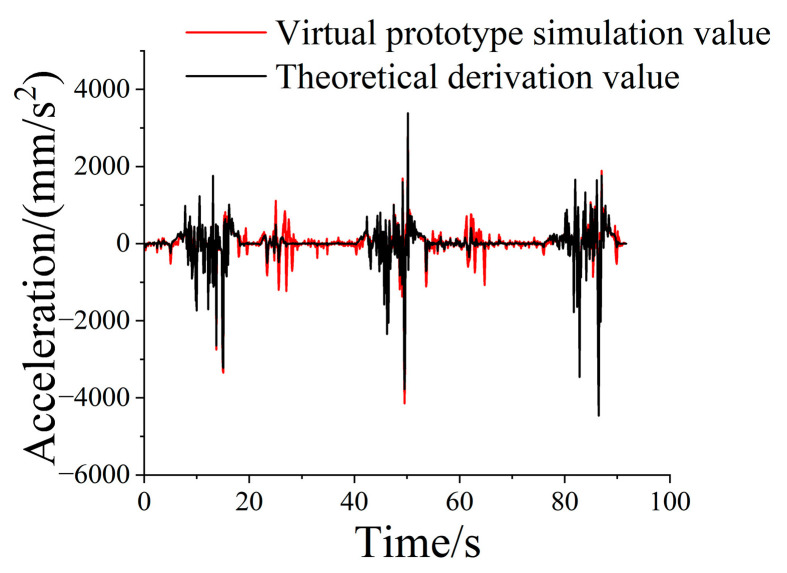
Comparison of simulated acceleration and theoretically derived acceleration.

**Figure 11 sensors-25-05585-f011:**
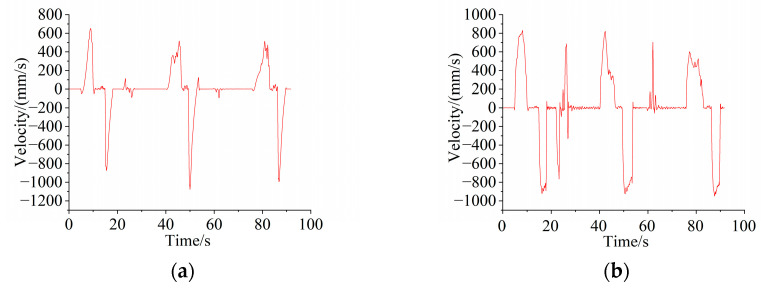
Velocity curve of marker point: (**a**) horizontal velocity; (**b**) vertical velocity.

**Figure 12 sensors-25-05585-f012:**
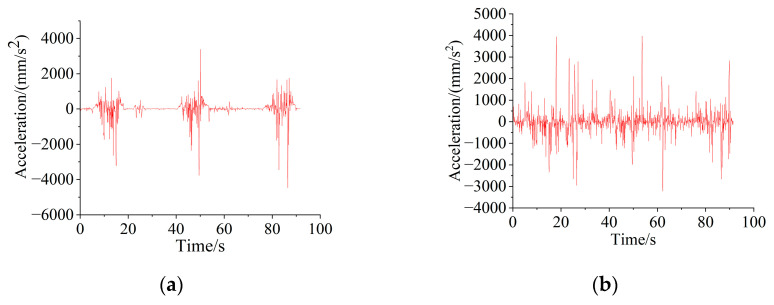
Acceleration curve of marker point: (**a**) acceleration in the horizontal direction; (**b**) acceleration in the vertical direction.

**Figure 13 sensors-25-05585-f013:**
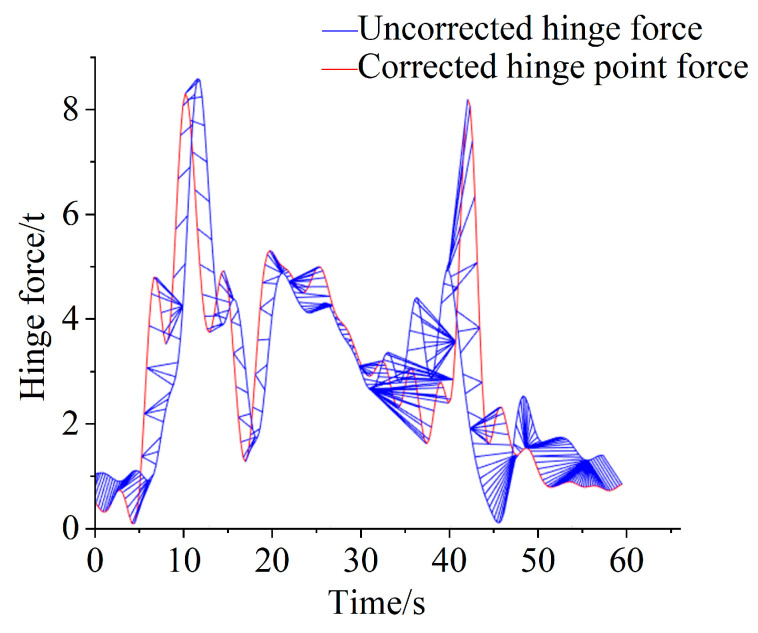
Comparison before and after the correction of the hinge point force.

**Figure 14 sensors-25-05585-f014:**
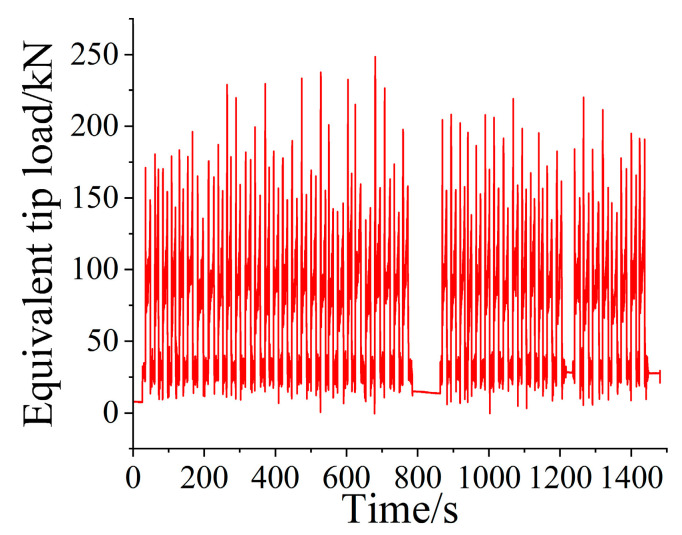
Bucket equivalent tip load spectrum.

**Figure 15 sensors-25-05585-f015:**
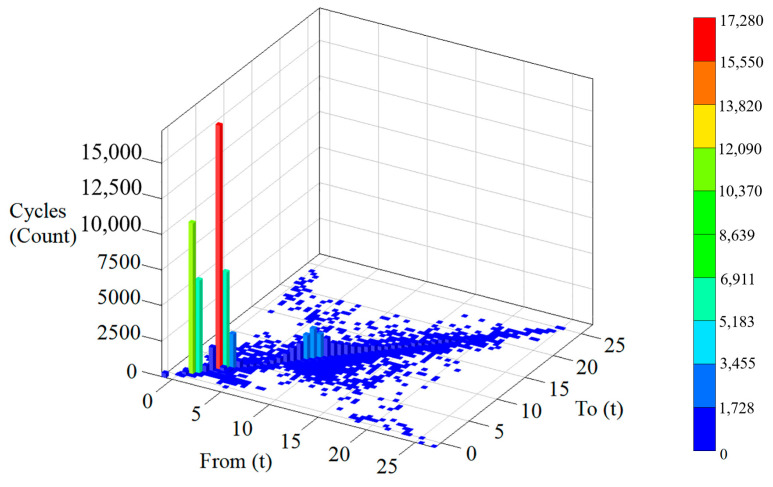
Rainflow count history after extrapolation.

**Figure 16 sensors-25-05585-f016:**
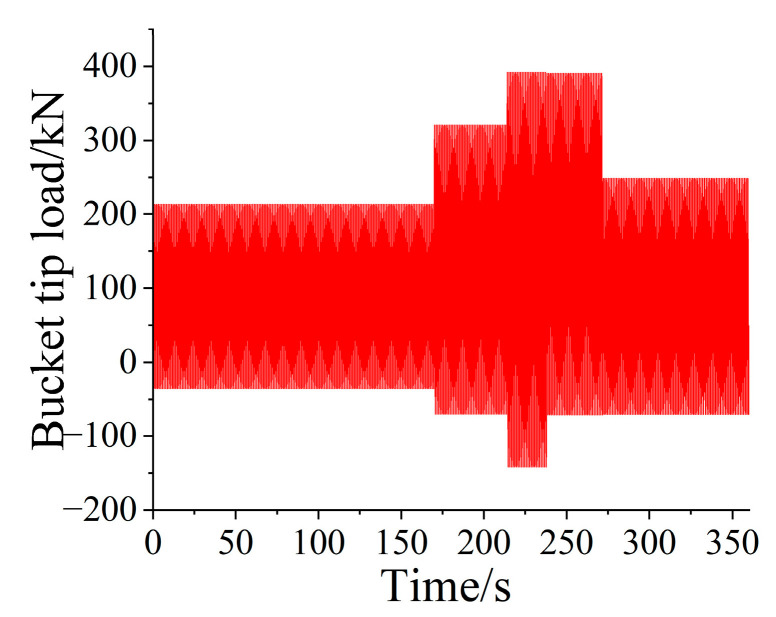
One-dimensional loading spectrum of bucket tip.

**Figure 17 sensors-25-05585-f017:**
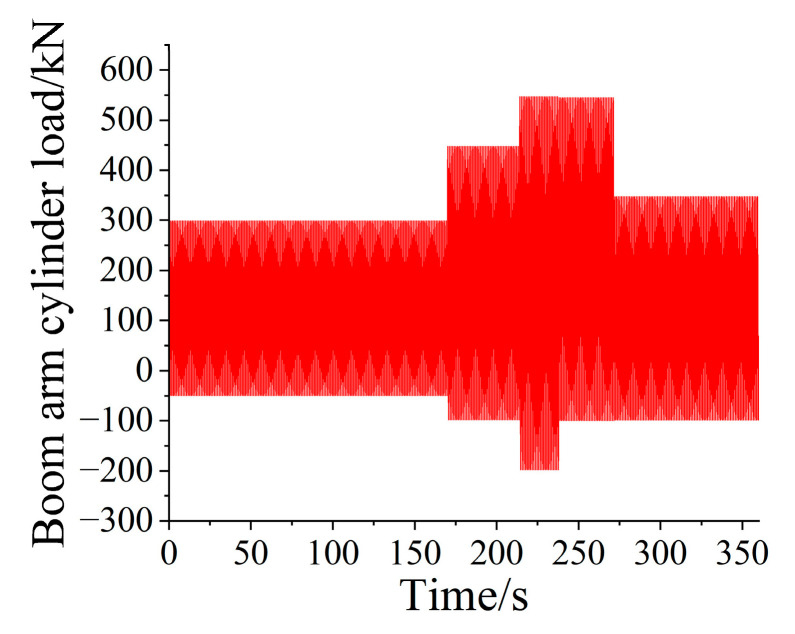
Loading spectrum of boom cylinder pressure program.

**Figure 18 sensors-25-05585-f018:**
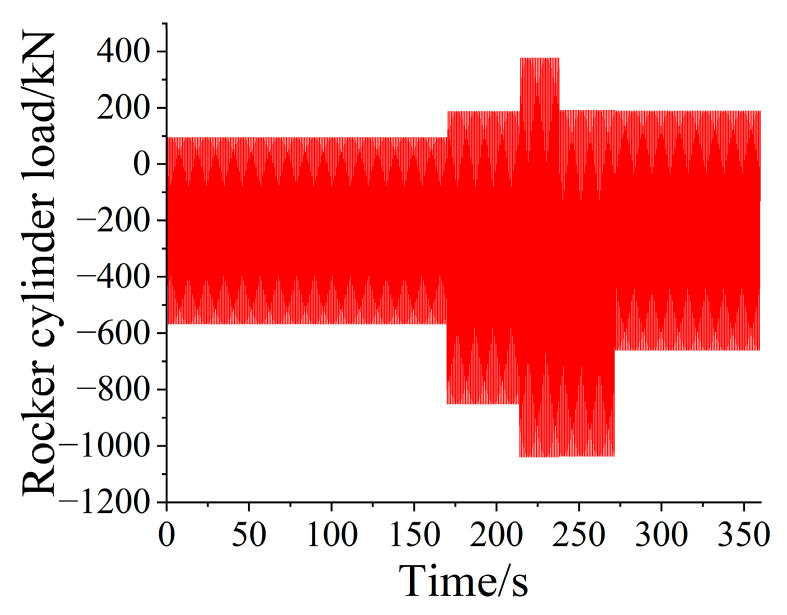
One-dimensional programmed load spectrum of rocker arm.

**Figure 19 sensors-25-05585-f019:**
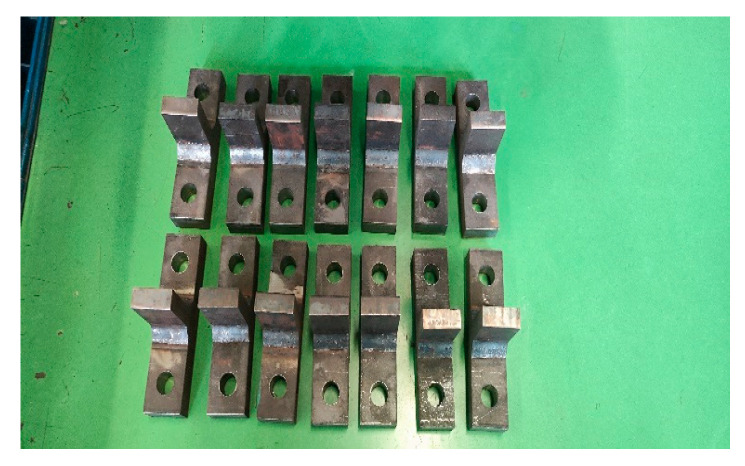
Weldment sample.

**Figure 20 sensors-25-05585-f020:**
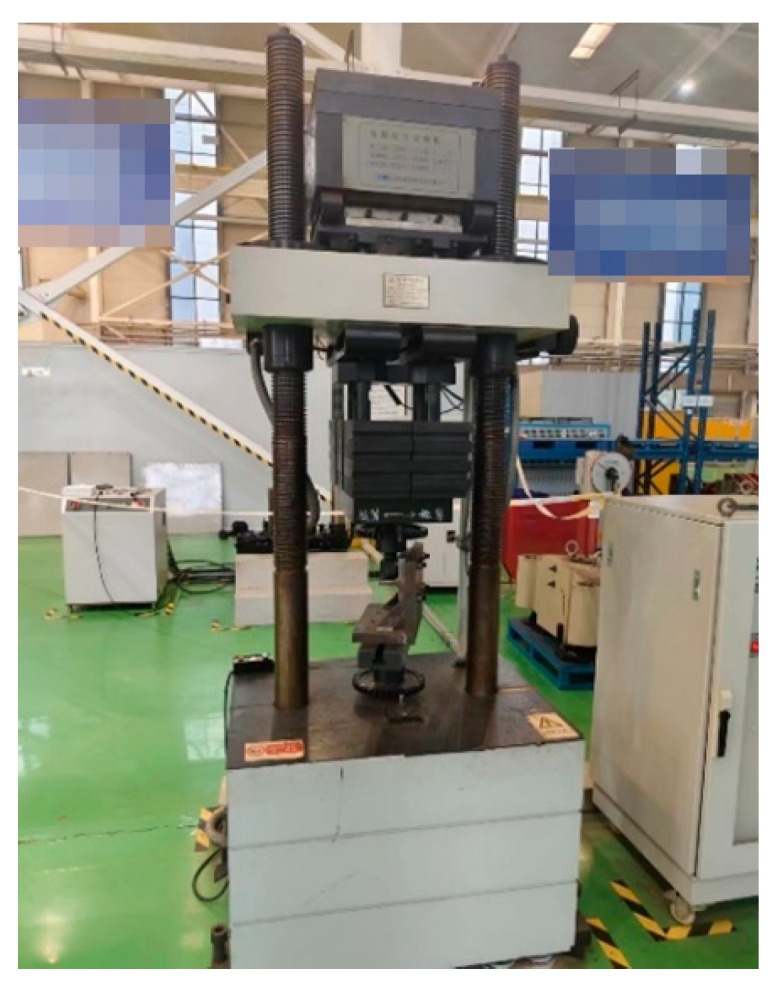
Fatigue test tooling physical drawing.

**Figure 21 sensors-25-05585-f021:**
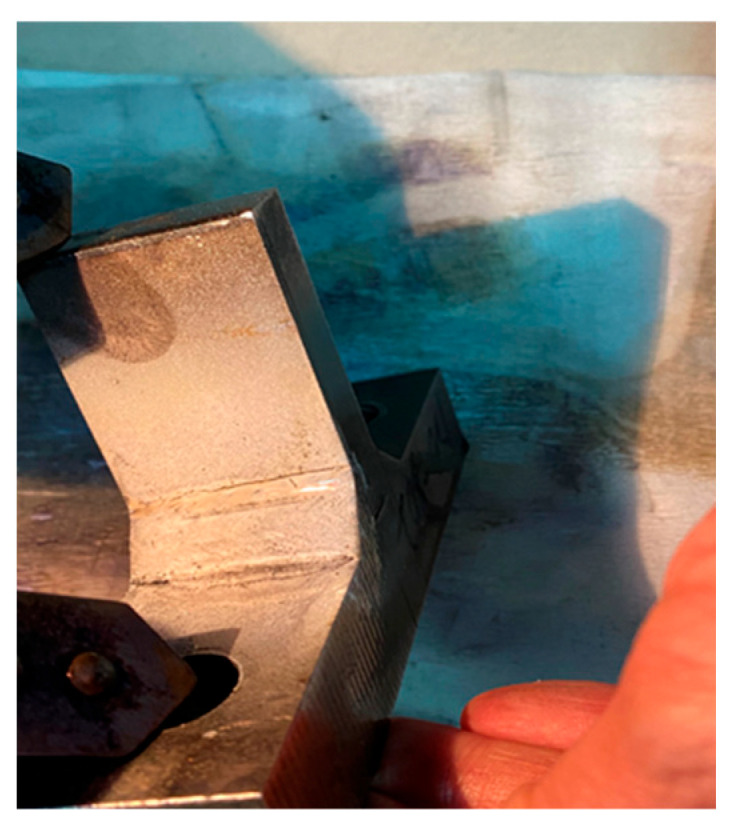
Fatigue failure diagram of the specimen.

**Figure 22 sensors-25-05585-f022:**
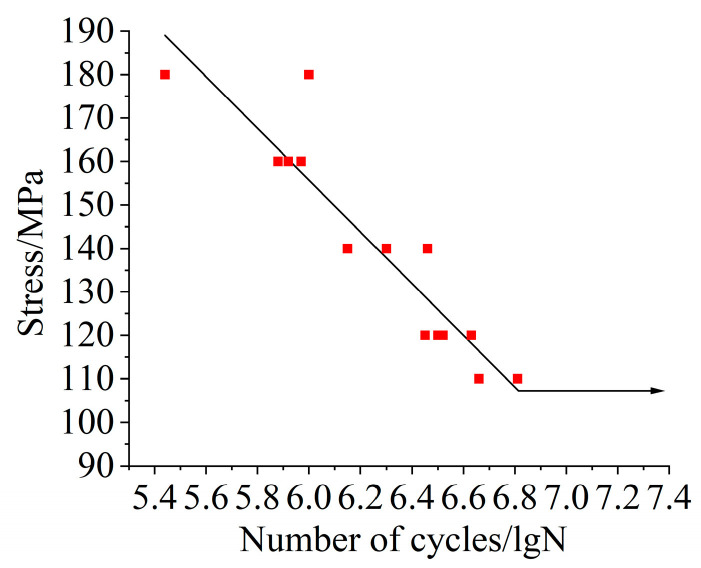
The S-N curve of Q345.

**Figure 23 sensors-25-05585-f023:**
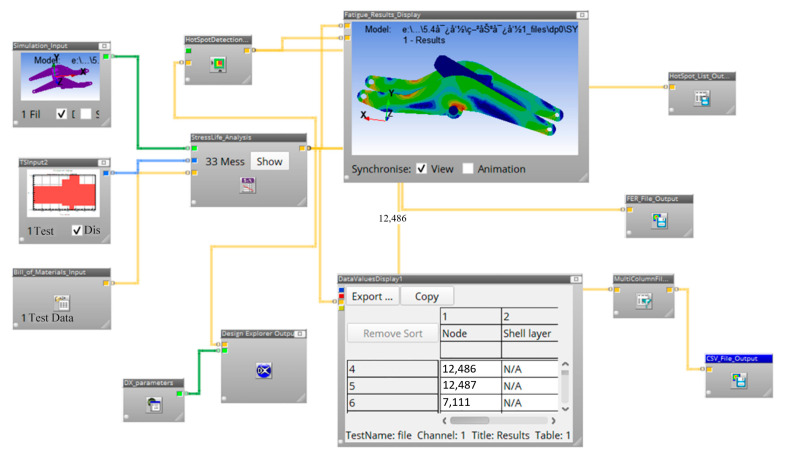
Flow chart of the fatigue simulation of the nCode boom.

**Figure 24 sensors-25-05585-f024:**
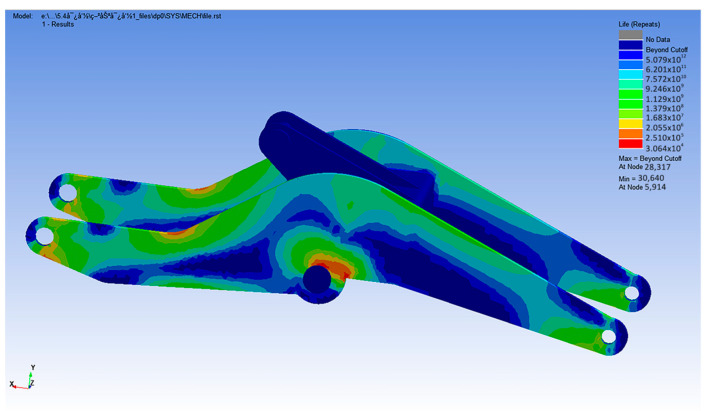
Fatigue life simulation diagram of the boom.

**Figure 25 sensors-25-05585-f025:**
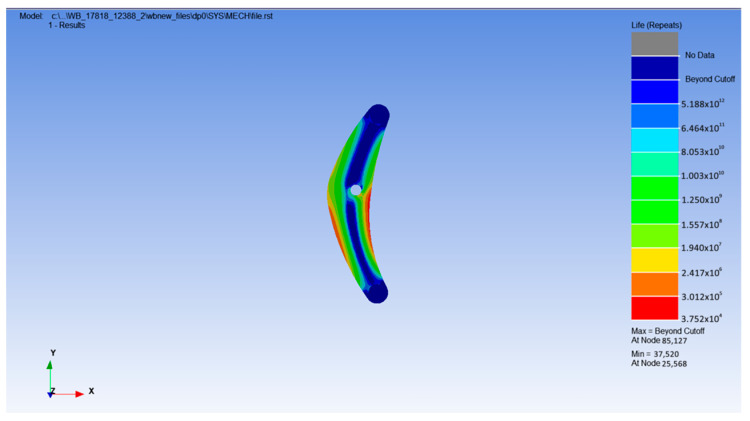
Fatigue life simulation diagram of the rocker arm.

**Table 1 sensors-25-05585-t001:** Load spectrum of 8 levels.

Load Class	Mean/kN	Amplitude/kN	Frequency
1	50	16	2106
2	50	50	2448
3	50	84	456
4	86	117	170
5	86	151	93
6	117	184	44
7	150	217	34
8	117	251	24

**Table 2 sensors-25-05585-t002:** Static mechanical properties of Q345.

Project	Performance Metric Parameters
Material	Q345
Tensile strength/MPa	490–675
Yield strength/MPa	345
Average tensile strength/MPa	583
Modulus of elasticity E/GPa	206
Type of steel	Low-alloy high-strength structural steel
Poisson’s ratio	0.27–0.30

**Table 3 sensors-25-05585-t003:** Static mechanical properties of Q345.

Stress/MPa	Number of Cycles/lgN	Average of Cycles	Standard Deviation of Cycles
180	5.44	5.72	0.28
180	6.00		
160	5.88	5.92	0.04
160	5.92		
160	5.97		
140	6.15	6.30	0.13
140	6.46		
140	6.30		
120	6.52	6.53	0.07
120	6.45		
120	6.63		
120	6.50		
110	6.81	6.74	0.08
110	6.66		

## Data Availability

The original contributions presented in this study are included in the article. Further inquiries can be directed to the corresponding author.

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
