# Peer review of "Compilation of Load Spectrum of Loader Working Device and Application in Fatigue Life Prediction"

_sensors, 2025, doi:10.3390/s25175585_

Round 1
Reviewer 1 Report
Comments and Suggestions for Authors
Comments and Suggestions for Authors:
- In Chapter One, there is a lack of research progress on the compilation of load spectrum. It is suggested to add an introduction to the research progress on the compilation of load spectrum.
- It is mentioned that “an automated detection and elimination process for abnormal spike data points was implemented within the nCode GlyphWorks software platform”. It is suggested to add a detailed explanation of the process.
- There are some problems in Section 3.1, such as “force and ” and “pull , ”. It is suggested to modify.
- There are problems with the annotations in the figures. For example, F1 is missing in Figure 5 and F1 is wrongly marked as F2 in Figure 6. It is suggested to modify.
- There are some problems with the formula in Section 3.1, such as equations(1). It is suggested to check the correctness and make corrections. There are still some minor errors, such as the representation of , the explanations of and , and and are not marked in the figure.
- The errors of the measured value and mapped value in Figure 8 are only simple statements. It is suggested to use error indicators to conduct error analysis on the measured value and mapped value.
- Some of the figures in the text, such as Figures 8 and 10 are not clear enough. It is suggested to increase the clarity.
- Section 3.3 mentions that the low amplitude load elision method was used to remove small loads below the fatigue limit of 60%. It is suggested to provide relevant evidence.
- In Section 4.1, it is suggested to explain why Q345 was chosen and whether it is consistent with the actual materials used in the loader.
- Figure 22 shows the fitting of the S-N curve of Q345. It is suggested to describe the detailed fitting method.
- In the end of Chapter 4, it is mentioned that the prediction results are highly consistent with the actual situation, and it is not rigorous enough to rule out other factors.It is suggested that the actual service cycle of 5-10 years be specified only for fatigue failure cases.
Reviewer 2 Report
Comments and Suggestions for Authors
This paper presents a study on fatigue life prediction for a wheel loader. While the topic is relevant and the work is interesting, there are several issues that need to be addressed:
- Please expand the literature review in the Introduction section. In particular, include more relevant studies on condition monitoring and prognostics, such as Enhanced particle filter and cyclic spectral coherence based prognostics of rolling element bearings; Sensorless robust anomaly detection of roller chain systems based on motor driver data and deep weighted KNN; Remaining useful life prediction combining advanced anomaly detection and graph isomorphic network.
- Clearly articulate the motivation behind your study and summarize your key contributions using bullet points to enhance readability.
- Include a section discussing the limitations of your current research and propose directions for future work to improve the completeness of the study.
Reviewer 3 Report
Comments and Suggestions for Authors
The manuscript presents a well-structured and technically sound approach to the load spectrum construction and fatigue life prediction of loader working devices. The authors integrate sensor-based data acquisition, finite element modeling, and fatigue theory within an industrially relevant context. However, some clarifications and enhancements would increase the manuscript’s clarity and rigor, particularly in terms of figure quality, validation range, and broader discussion.
1.Please clarify what distinguishes this study from prior works in load spectrum modeling.
2.Some figures are dense and unclear, please enhance resolution and labeling, e.g. figure 4, 8, 14, 17.
3.Clarify how extrapolation coefficients were determined in Section 3.3.
4.The S-N curve in Figure 22 should include error ranges or standard deviation.
5.Please discuss how variability in field conditions may influence model applicability.
6.Provide more evidence or explanation for the accuracy of the inertial correction model.
7.Please add units and enhance layout for Table 1.
8.Briefly compare this approach with other commonly used fatigue prediction techniques.
9.Consider discussing how AI or sensor fusion might improve prediction in future work.
10.Include more quantitative summary in the conclusion to reinforce findings.
Round 2
Reviewer 1 Report
Comments and Suggestions for Authors
The authors have made revisions to the suggestions I put forward, and I have no further suggestions.